# Humoral and T Cell Immune Responses against SARS-CoV-2 after Primary and Homologous or Heterologous Booster Vaccinations and Breakthrough Infection: A Longitudinal Cohort Study in Malaysia

**DOI:** 10.3390/v15040844

**Published:** 2023-03-25

**Authors:** Jolene Yin Ling Fu, Muhammad Harith Pukhari, Maria Kahar Bador, I-Ching Sam, Yoke Fun Chan

**Affiliations:** 1Department of Medical Microbiology, Faculty of Medicine, University of Malaya, Kuala Lumpur 50603, Malaysia; jolenefyl@gmail.com (J.Y.L.F.); mhbpbs@gmail.com (M.H.P.); mariakb@ummc.edu.my (M.K.B.); jicsam@ummc.edu.my (I.-C.S.); 2Department of Medical Microbiology, University Malaya Medical Centre, Kuala Lumpur 59100, Malaysia

**Keywords:** SARS-CoV-2, Omicron, BNT162b2 vaccine, ChAdOx1-S vaccine, booster, neutralizing antibody, T cell responses

## Abstract

Vaccine efficacy against SARS-CoV-2 could be compromised by the emergence of SARS-CoV-2 variants and it is important to study how it impacts the booster vaccination regime. We investigated the humoral and T cell responses longitudinally in vaccinated uninfected (*n* = 25) and post-COVID-19 individuals (*n* = 8), and those who had received a BNT162b2 booster following complete two-doses regimes of either BNT162b2 (homologous) (*n* = 14) or ChAdOx1-S (heterologous) (*n* = 15) vaccines, by means of a SARS-CoV-2 pseudovirus neutralization test and QuantiFERON SARS-CoV-2 assay. Vaccinated post-COVID-19 individuals showed higher neutralizing antibodies with longer durability against SARS-CoV-2 wild type (WT) and Omicron spikes, but demonstrated similar declining T cell responses compared to the uninfected vaccinated. Two doses of BNT162b2 induced higher neutralizing antibodies against WT and T cell responses than ChAdOx1-S for six months. The BNT162b2 booster confers a greater humoral response against WT, but a similar cross-neutralizing antibody against Omicron and T cell responses in the homologous booster group compared to the heterologous booster group. Breakthrough infection in the homologous booster group (*n* = 11) significantly increased the neutralizing antibody, but T cell responses remained low. Our data may impact government public health policy regarding the administration of mix-and-match vaccines, where both vaccination regimes can be employed should there be shortages of certain vaccines.

## 1. Introduction

The SARS-CoV-2 Omicron (B.1.1.529) variant of concern (VOC) was first reported in South Africa in November 2021 and has since spread globally, rapidly replacing Delta as the dominant VOC during the coronavirus disease 2019 (COVID-19) pandemic [1]. More than 30 of the 50 mutations in the Omicron genome occur in the spike region alone, giving the virus the advantage of increased infection and transmission [1,2]. Furthermore, Omicron can escape vaccine-induced neutralizing antibodies, with two doses of mRNA, adenoviral vector and whole inactivated virus vaccines shown to induce minimal antibody responses against Omicron [3,4]. As COVID-19 vaccine-induced immunity wanes over time, a third vaccine (booster) dose has been introduced to boost immune response against SARS-CoV-2. Although the booster dose increases cross-neutralizing antibodies against Omicron, the antibody levels remain several folds lower than antibodies against wild-type (WT) spikes [5]. In contrast to antibodies, the T cell response is more stable as it can cross-recognize Omicron [6], and remains detectable despite low levels or a complete absence of detectable antibodies following mild symptomatic or asymptomatic COVID-19 infection [7]. 

While many studies provided insight into the immunological responses following vaccination and SARS-CoV-2 infection, most were carried out in healthcare workers, with convenience samples or for a short duration [4,8,9,10]. Few studies, especially in Asia [11,12,13], have characterized the correlation between antibody and T cell responses within the same individual over time, despite evidence of heterogeneous immune responses between individuals and that antibodies and T cells act together at different stages of SARS-CoV-2 infection or vaccination to protect against severe diseases and reinfection [14,15]. Therefore, it is crucial to evaluate the neutralizing antibody and T cell responses induced by primary and booster vaccinations against Omicron, especially in people with different levels of preexisting immunity, including those with or without breakthrough infections.

Malaysia used several COVID-19 vaccines, including mRNA (Pfizer–BioNTech BNT162b2 [BNT]), adenoviral vector (Oxford-AstraZeneca ChAdOx1-S [ChAd] and CanSino), and whole inactivated virus (CoronaVac) vaccines. As of 31 December 2022, a total of 72,326,604 doses has been administered, with the BNT vaccine accounting for 61.9% and ChAd accounting for 7.9% [16]. Malaysia provided the booster dose as part of its vaccination campaign beginning September 2021 [17], with an approximate uptake of 49.9% as of 31 January 2023 [18]. Due to the limited availability of vaccines at the time and emerging studies on improved heterologous vaccine efficacy [19,20], a unique situation was created in which homologous and heterologous vaccination regimes, with BNT as the preferred booster regardless of the primary course, were applied to rapidly increase the booster coverage among Malaysians [21]. 

In this study, we provide a side-by-side longitudinal analysis of neutralizing antibody and T cell responses for up to 3 months in a cohort of 73 individuals, including uninfected and post-COVID-19 individuals receiving two doses of vaccines, and those who received a BNT booster dose following complete two-dose regimes of either BNT (homologous booster group) or ChAd (heterologous booster group) vaccines. The immune responses of patients in the homologous booster group who did (homologous booster breakthrough group) or did not (homologous booster non-breakthrough group) experience COVID-19 infections after the booster were also compared. An assessment of the impact of booster vaccination on immune responses to SARS-CoV-2, especially the VOCs, provides important insights for policy decisions involving COVID-19 resources, mitigation measures and future vaccine development.

## 2. Materials and Methods

### 2.1. Study Design

In this longitudinal study, 100 individuals were recruited between 26 August 2021 and 8 March 2022, and 27 individuals were excluded from the study due to missed time-points, incomplete data or positive for SARS-CoV-2 infection midway through the study (Appendix A). The individuals were divided into 5 groups: (1) 25 previously uninfected individuals receiving two primary doses of the BNT vaccine; (2) 8 post-COVID-19 individuals receiving two primary doses of vaccine (enrolled in the study irrespective of which vaccine they received); (3) 14 BNT-primed uninfected individuals receiving BNT booster (homologous booster group); (4) 15 ChAd-primed uninfected individuals receiving BNT booster (heterologous booster group); and (5) 11 individuals who experienced COVID-19 breakthrough infections after receiving three doses of BNT vaccines (homologous booster breakthrough group). Post-COVID-19 individuals were identified based on laboratory confirmation of previous SARS-CoV-2 infection. Breakthrough infection was confirmed via detection of antibodies against nucleoprotein of SARS-CoV-2 using Elecsys Anti-SARS-CoV-2 (Roche Diagnostics International Ltd, Rotkreuz, Switzerland). Power analysis was not performed on the number of individuals due to the selection criteria and voluntary basis of participation.

For uninfected and post-COVID-19 individuals receiving two doses of vaccines, given 21 days apart, plasma was collected immediately before the first vaccine dose (T1), and 21 days (T2) and 3 months (T3) after the second vaccine dose. For the booster dose, plasma was collected before booster (B1), and 21 days (B2) and 3 months after booster (B3). An additional time-point (B4) was collected 6 months after booster for the homologous booster group. Demographics of patients are depicted in Appendix A. 

### 2.2. Plasma Isolation

Venous blood was collected in a 10 mL EDTA tube (for antibody assays) and heparin tube (for interferon-γ [IFN-γ] release assay). The EDTA blood tube was centrifuged at room temperature for 15 min at 2000× *g*. The serum samples were aliquoted and stored at −20 °C.

### 2.3. Elecsys Anti-Spike and Anti-Nucleoprotein Immunoassay

All serum samples were tested for antibodies against receptor binding domain (RBD) of spike protein and nucleoprotein of SARS-CoV-2 using Elecsys Anti-SARS-CoV-2 S (Elecsys anti-S, Cat # 09289275190) and Elecsys Anti-SARS-CoV-2 (Elecsys anti-N, Cat # 09203079190) (Roche Diagnostics), respectively. All samples were processed according to manufacturer’s instructions. Briefly, 200 µL of serum was transferred to a sample cup before being loaded into the cobas e 801 module. Serum collected from individuals prior to receiving the first dose of vaccine was tested undiluted while those collected after the second or booster dose were diluted 50× using Diluent Universal reagent (Roche Diagnostics). 

The results by Elecsys anti-S have a cut-off point of ≥0.80 U/mL to differentiate samples as reactive and non-reactive. The Elecsys anti-N is a semi-quantitative assay, with cutoff index (COI) <1.0 considered as non-reactive, and ≥1.0 considered reactive. All data on anti-S and anti-N are reported in Appendix A. 

### 2.4. Cells and Pseudoviruses

Human embryonic kidney cells (HEK-293T, ATCC CRL-1573, ATCC, Manassas, VA, USA) and HEK-293T-ACE2-TMPRSS2-mCherry (NR-55293, BEI Resources, NIAID/NIH, Manassas, VA, USA) were grown in Dulbecco’s minimal essential medium (Gibco, Waltham, MA, USA) with 10% (*v*/*v*) FBS, 1% (*v*/*v*) penicillin-streptomycin (P/S), 2 mM (*v*/*v*) of L-glutamine, and 1% (*v*/*v*) sodium pyruvate at 37 °C with 5% CO_2_. To generate SARS-CoV-2 pseudovirus, HEK-293T cells were transfected with pTwist SARS-CoV-2 spike WT expression plasmid (#164436, Addgene, Watertown, MA, USA) encoding a codon-optimized SARS-CoV-2 spike gene with an 18-residue truncation in the cytoplasmic tail, packaging plasmid (psPAX) (#12260, Addgene) and luciferase reporter plasmid (pLenti CMV PURO LUC) (#17477, Addgene). Forty-eight hours after transfection, the supernatant containing the SARS-CoV-2 pseudovirus was collected and clarified by centrifugation. A pseudovirus bearing Omicron (B.1.1.529) SARS-CoV-2 spike (#179907, Addgene) was generated using a similar method.

### 2.5. Pseudovirus Neutralization Assay

All sera were heat-inactivated for 30 min at 56 °C before use in the neutralization assay. HEK-293T-ACE2-TMPRSS2-mCherry cells were seeded in a 96-well plate a day before the neutralization assay. Two-fold serially diluted serum samples, ranging from 1:10 to 1:320, were mixed with 1.5 × 10^5^ relative luminescence units (RLU) of pseudovirus and incubated for 1 hr at 37 °C. Sera from post-COVID-19 patients (collected during the Omicron predominant wave) was used as a positive control. The media in the HEK-293T-ACE2-TMPRSS2-mCherry cells was replaced with 50 µL of serum–virus mixture in duplicate and incubated for 48 hrs at 37 °C with 5% CO_2_. A volume of 40 µL of Bright-GLO luciferase substrate (Promega, Madison, WI, USA) was added to each well and the luminescence signal was measured using a Glomax multi-detection plate reader (Promega). The sera dilution that led to 50% reduction of RLU compared to control was calculated using a nonlinear regression curve fit with GraphPad Prism v5.1 (GraphPad Software, San Diego, CA, USA) and reported as neutralizing antibody titers (IC_50_). The neutralization assay against WT and Omicron pseudoviruses were tested in parallel to facilitate side-by-side comparison in the correlation studies. An IC_50_ below 1/10 serum dilution was considered negative and designated a value of 5 for statistical analysis. The neutralization results are tabulated in Appendix A.

### 2.6. Interferon-γ Release Assay

To evaluate the T cell responses to vaccination, SARS-CoV-2-specific IFN-γ response was measured using QuantiFERON SARS-CoV-2 Starter Pack (# 626715, Qiagen, Hilden, Germany) and Extended Pack (# 626815, Qiagen). The Starter Pack consists of an Ag1 tube that contains CD4+ T cell epitopes from the S1 subunit (RBD) of spike protein and an Ag2 tube with both CD4+ and CD8+ T cell epitopes from the S1 and S2 subunits of spike protein. The Ag3 tube from the Extended Pack contains CD4+ and CD8+ T cell epitopes from S1 and S2 (as in Ag2), and additional immunodominant CD8+ T cell epitopes of the whole genome including spike, nucleoprotein and membrane. Briefly, whole blood collected in heparin tubes was incubated with the peptide pools or mitogen (positive control) for 16-24 h. The QuantiFERON tubes were centrifuged for 15 min at 2000× *g* and the serum was collected and stored at −20 °C. The IFN-γ response against Ag1, Ag2 and Ag3 was quantified by QuantiFERON ELISA (# 626410, Qiagen) and a cutoff value of 0.15 IU/mL was used to indicate a positive T cell-mediated immune response. The blood incubation in Ag1, Ag2 and Ag3 tubes, and ELISA assay were performed in parallel to facilitate side-by-side comparison in the correlation studies.

### 2.7. Statistical Analysis

The effect of the age and time interval from vaccination to sampling between groups was determined using unpaired two tailed T-test. Fisher’s exact test was used to test differences in sex and to compare positive rates between groups. The Wilcoxon matched-pairs signed rank test was used to determine the differences within groups between time-points and between Ag1, Ag2 and Ag3 at the same time-point. Comparison between the uninfected and post-COVID-19 groups at the same time-point were assessed using the Mann–Whitney U-test. All tests were two-sided and statistical significance was defined as a *p*-value < 0.05. The normality of the dataset was tested by means of the D’Agustino–Pearson test. Correlations between neutralizing antibody and T cell responses were assessed by the Spearman rank correlation coefficient (rho). All statistical tests were performed with GraphPad Prism.

## 3. Results

### 3.1. Vaccinated Post-COVID-19 Subjects Demonstrated Higher Neutralizing Antibody with Longer Durability but Similar Decline in T Cell Responses Compared to Vaccinated Uninfected Subjects

To compare the neutralizing antibody and T cell responses in uninfected and post-COVID-19 individuals receiving two doses of vaccines, plasma samples were collected before the first dose (T1), and 11–27 days (T2) and 63–89 days (T3) after the second dose (Figure 1A). For post-COVID-19 individuals, the time between COVID-19 diagnosis (pre-Omicron wave) and first plasma ranged between 56 to 136 days. The post-COVID-19 group has a significantly higher median age (*p* = 0.0071) compared to the uninfected group. 

The majority (84%, 21/25) of uninfected vaccinated individuals demonstrated an increase (*p* < 0.0001) in neutralizing antibody against WT within 21 days after the second dose, but they were near the lower limit of detection against Omicron, with GMT 18.4-fold (*p* < 0.0001) lower than WT (Figure 1B). Antibody against WT waned significantly (*p* < 0.0001) in the following three months, while the antibody activity against Omicron remained near the detection limit and 6.1-fold (*p* = 0.0002) lower than WT. Prior to vaccination, three of the eight post-COVID-19 individuals retained low levels of neutralizing antibody against WT, and only one of the three positives had detectable antibodies against Omicron. Two doses of vaccines induced high neutralizing antibodies against WT (*p* = 0.0076) and Omicron (*p* = 0.0156), and while the antibody against Omicron remained stable after three months, the antibody activity against WT dropped 2.8-fold (*p* = 0.0156). We observed that vaccinating post-COVID-19 individuals induced significantly higher neutralizing antibodies against both WT and Omicron across all time points compared to the vaccinated uninfected group (Figure 1C). 

Next, we evaluated the T cell responses by measuring the level of IFN-ɣ released by CD4+ when stimulated against spike S1 subunit (Ag1), CD4+ and CD8+ against spike S1 and S2 subunits (Ag2), and CD8+ against immunodominant epitopes of the whole proteome coupled with Ag2 components (Ag3). CD4+ and CD8+ responses could be detected in all individuals in both groups within the first 21 days after the second dose, with the post-COVID-19 group showing significantly higher CD4+ response against Ag1 (median 3.4, IQR 0.7–6.6) compared to the uninfected group (0.7 IU/mL, IQR 0.3–1.8, *p* = 0.0357; Figure 1D). CD4+ and CD8+ responses declined to similar levels in both groups after three months, and remained detectable at low levels with medians ranging between 0.2–0.5 IU/mL (IQR 0.2–0.7) for uninfected and 0.7–1.4 IU/mL (IQR 0.1–4.5) for the post-COVID-19 group. Regardless of infection status, Ag2 (*p* = 0.0013) and Ag3 (*p* = 0.0009) stimulated significantly greater responses than Ag1 for up to 3 months after the second dose. The post-COVID-19 group demonstrated stronger responses against Ag3 than Ag2 (*p* = 0.0078) after three months. 

The uninfected group generally demonstrated a weak negative correlation between the neutralizing antibody and T cell responses that persisted over time after vaccination, with rho values ranging between −0.03 to 0.10 (Appendix A). The correlation was much stronger in the vaccinated post-COVID-19 group, with rho values ranging between 0.28 to 0.57 (Appendix A). 

### 3.2. The Homologous and Heterologous Booster Regime Induced Comparable Neutralizing Antibody and T Cell Responses

For the homologous booster group, plasma was collected before the booster dose (B1), 181–211 days (median 199 days) after the second dose (Figure 2A). For the heterologous booster group, plasma (B1) was collected 110–180 days (median 153 days) after the second dose. For subsequent time-points for both groups, plasma was collected 17–34 days (B2) and 80–111 days (B3) after the booster. An additional time-point for the homologous booster group (B4) was collected 170–200 days after the booster. The homologous booster group median age is significantly (*p* = 0.0129) higher than the heterologous booster group. 

The majority (71%, 10/14) of the BNT-primed individuals retained low levels of neutralizing antibody against WT 6 months after second dose, but they were near detection limit against Omicron, with GMT 6.6-fold (*p* = 0.002) lower than WT (Figure 2B). The booster induced high neutralizing antibody against WT (*p* = 0.0001) and a moderate increase against Omicron that was 5.1-fold (*p* = 0.0001) lower than WT. Both antibodies waned significantly in the following three months, with neutralizing antibodies against Omicron dropping 6.6-fold (*p* = 0.0001) compared to WT after six months. Notably, the majority (93%, 14/15) of the ChAd-primed individuals lost detectable neutralizing antibodies against both WT and Omicron prior to the booster. Boosting the ChAd-primed group initially induced high neutralizing antibodies (*p* < 0.0001) against WT, but not Omicron, which was 7.3-fold (*p* < 0.0001) lower than WT. Neutralizing antibodies against WT and Omicron both dropped significantly after three months. Initially, the booster appears to induce a better antibody response against Omicron in the homologous booster group compared to heterologous booster group (Figure 2C). After three months, neutralizing antibodies against Omicron dropped to similar levels in both groups, while neutralizing antibodies against WT remained significantly higher in the homologous booster group. 

T cell responses could be detected in 79% (11/14) of BNT-primed individuals before the booster, with significantly higher CD4+ and CD8+ responses against Ag1 (*p* = 0.0005), Ag2 (*p* = 0.0017) and Ag3 (*p* = 0.015) compared to the ChAd-primed group (47%, 7/15) (Figure 2D). The booster initially induced robust T cell responses in the homologous and heterologous booster groups against Ag1, Ag2, and Ag3. After three months, the heterologous booster group showed significant waning of T cell responses against Ag1 (median 0.1, IQR 0–0.4 IU/mL, *p* = 0.0021), Ag2 (median 0.1, IQR 0–0.6 IU/mL, *p* = 0.0277) and Ag3 (median 0.1, IQR 0–0.6 IU/mL, *p* = 0.0014), with only 47% (7/15) having detectable T cell responses compared to 79% (11/14) in the homologous booster group. Although the homologous booster group demonstrated significant decline in T cell responses against Ag3 (*p* = 0.0052) after six months, their responses remained higher compared to the heterologous booster group at 3 months post-booster. Regardless of vaccination regimes, Ag2 stimulated better T cell responses than Ag1 following the booster (Appendix A). 

The correlation between neutralizing antibody and T cell responses was most robust in the BNT-primed group before the booster, with rho values ranging between 0.55 and 0.75 (Appendix A). The correlation remained moderate for up to 6 months, with rho values dropping slightly to 0.35–0.53. The ChAd-primed group showed negative correlation against Ag1 (rho −0.13), and moderate correlation against Ag2 (rho 0.44) and Ag3 (0.32) before the booster (Appendix A). The correlation remained weak (rho −0.08–0.26) for up to three months after the booster, with the exception of Ag3 (rho 0.79). 

### 3.3. Low T Cell Responses upon Breakthrough Infection in the Homologous Booster Group 

Eleven individuals from the homologous booster group experienced breakthrough infections between 3 months and 6 months post-booster (Figure 2A). All COVID-19 infections were laboratory-confirmed and occurred within 3–35 days after B3 time-point collection, which was 111–141 days post-booster. The B4 sample was collected at 61–93 days after breakthrough infection. All individuals with breakthrough infections were either asymptomatic or experienced mild COVID-19 symptoms (WHO ordinal scale of 0–2) [22].

The homologous booster breakthrough group demonstrated comparable neutralizing antibodies against WT and Omicron as the non-breakthrough group prior to breakthrough infection (Figure 2B). The booster significantly induced neutralizing antibodies against WT (*p* = 0.001) and Omicron (*p* = 0.001) after 21 days, but Omicron remained significantly lower than WT (5.2-fold, *p* = 0.001). The antibody activity also waned rapidly in the following three months, with complete absence of neutralizing antibodies against Omicron in two individuals after three months. Following breakthrough infection, neutralizing antibodies increased significantly against WT (*p* = 0.001) and Omicron (*p* = 0.001). 

A majority (82%, 9/11) of the homologous booster breakthrough group demonstrated similar T cell responses as the non-breakthrough group prior to the breakthrough infection (Figure 2D). The booster significantly induced T cell responses against Ag1 (median 0.5, IQR 0.1–1.7 IU/mL, *p* = 0.0322), Ag2 (median 1.0, IQR 0.1–1.9 IU/mL, *p* = 0.0098), and Ag3 (median 0.7, IQR 0.2–2.3 IU/mL, *p* = 0.0137), but responses decreased significantly against Ag2 (median 0.4 IQR 0–1.3 IU/mL, 0.042) three months later. Surprisingly, T cell responses in one (9%) of the eleven breakthrough infections could not be detected after the breakthrough infection, with levels comparable to the non-breakthrough group at 6 months post-booster. Like the non-breakthrough group, the breakthrough group demonstrated greater T cell responses against Ag2 compared to Ag1 following the booster, and remained significantly higher after the breakthrough infection (Appendix A). The T cell responses against Ag3 were also significantly higher than Ag1 before and after the breakthrough infection.

Before the booster, the correlation between the neutralizing antibody and T cell responses in the homologous booster breakthrough group was weak to moderate, with the rho number ranging between 0.21 to 0.47 (Appendix A). The homologous booster breakthrough group demonstrated a negative correlation (rho −0.49 to −0.53) 21 days after the booster, which only turned positive (rho number 0.19–0.30) after three months. Following the breakthrough infection, the correlation (rho 0.38–0.60) remained similar to the non-breakthrough group.

## 4. Discussion

We observed that two doses of vaccines are insufficient to induce neutralizing antibodies against Omicron in uninfected individuals while vaccinated post-COVID-19 individuals showed high neutralizing antibodies against both WT and Omicron. Our data are similar to earlier reported results that neutralizing antibodies induced by two doses of BNT vaccines in uninfected individuals waned rapidly within months after vaccination and are insufficient to stimulate a significant humoral response against Omicron [5,23,24,25]. Likewise, the lack of neutralizing antibodies in the post-COVID-19 group prior to vaccination suggests that the neutralizing antibody induced by previous infection waned rapidly [25], though memory B-cell responses have been shown to remain stable for up to 6 months in vaccinated and post-COVID-19 individuals [26]. Therefore, re-exposure to virus antigens such as vaccinating previously infected individuals, further boosters and breakthrough infections, as observed in our study, could expand the repertoire of memory B cell clones expressing potent antibodies over a longer period of time [11,27] and produce antibodies that are less susceptible to escape variants [28,29,30], which were absent in the double-dose vaccinated uninfected group [31,32].

We showed that the majority of the BNT-primed individuals retained a low level of neutralizing antibodies against WT despite having a longer gap (median 199 days vs. 153 days, *p* < 0.0001) in between the second dose and sample collection time compared to ChAd-primed group. The greater vaccine efficacy conferred by two doses of BNT was similarly reflected in other studies [33,34], though the time point measured was shorter. The data on the heterologous vaccine regime vary depending on the number of doses, with two doses of the heterologous vaccine (ChAd/BNT) shown to be more effective compared to the homologous vaccine (BNT/BNT and ChAd/ChAd) [24,33,34,35,36], while three doses of the heterologous and homologous vaccine demonstrated a similar vaccine effectiveness and induced comparable antibody responses against Omicron [26,37], which was similarly observed in our study. One caveat in this regard is that we did not analyze the time points beyond 3 months for the heterologous booster group. The much reduced antibody activity against Omicron observed in this study is indicative of a higher degree of immune evasion, and since neutralizing antibody levels are associated with protection against infection [38], this implies that vaccine efficacy against Omicron infection decreases over time [39]. 

Although vaccine efficacy against symptomatic infections decreases over time, protection against progression to severe disease remains, likely due to persistent T cell responses [40]. We showed that two doses of vaccines induced CD4+ and CD8+ responses in the majority of uninfected and post-COVID-19 individuals, but CD4+ and CD8+ responses in both groups subsequently declined to comparable levels after three months. A similar waning of T cell responses were also observed in a previous study, with IFN-ɣ levels typically ranging between 0.7–1.2 U/mL [41]. Given that all of the uninfected and the majority (87.5%) of the post-COVID-19 individuals were vaccinated with the BNT vaccine, an mRNA-based vaccine that encodes for the full length spike protein [42], our data indicated that the vaccine could robustly induce CD4+ and CD8+ responses against spike. The post-COVID-19 group demonstrated stronger CD8+ responses against immunodominant epitopes of the whole genome than spike alone, suggesting that other genome proteins such as nucleoproteins could induce robust CD8+ responses [43], thus mitigating the risk of immune escape of new variant strains. The higher CD4+ response in the vaccinated post-COVID-19 group compared to the uninfected group indicated that spike-specific T cell responses are dominated by CD4+ instead of CD8+ T cells, given that memory CD4+ T cells are primarily generated following infection [26,40].

We showed that the CD4+ and CD8+ responses in BNT-primed group remained higher than ChAd-primed group for at least 6 months after the second dose. These data reflect findings by Moore et al. [26], though other studies have also demonstrated that both vaccines induced similar T cell responses for up to 4 months [24,44]. The booster dose induced similar CD4+ and CD8+ responses in homologous and heterologous booster groups, and the magnitude of the responses in both groups following the booster dose were comparable to 21 days post second dose, indicating that T cell responses were not significantly boosted upon the third dose [45]. The T cell responses in the heterologous booster group waned rapidly after 3 months, possibly due to the limited expansion of spike-specific T cells [46]. Again, data on T cell responses induced by the heterologous vaccine regime differ based on the number of doses, with two doses of the heterologous vaccine shown to induce better T cell responses [47], while data from our study demonstrated that three doses of the heterologous vaccine regime produced T cell responses comparable to the homologous booster regime. Together with the antibody levels, our data showed that the homologous booster regime with BNT vaccine appears to provide a slight, but insignificant, advantage over the heterologous booster regime, consistent with other reports [48]. 

While protection against breakthrough infection requires a combination of high neutralizing antibody and T cell responses [49], our data showed that protection from breakthrough infection in the homologous booster group was not associated with any of the analyzed immune parameters. One possibility for the breakthrough infection was that the level of humoral and T cell response in our cohort was insufficient to protect against the Omicron variant, which was predominant at the time of sampling [50]. Despite a significant increase in neutralizing antibodies following the breakthrough infection, CD4+ and CD8+ responses were not boosted to levels observed in another study [49]. The lack of differences in both the CD4+ and CD8+ responses between the breakthrough and non-breakthrough group are likely influenced by a combination of factors including the severity of the breakthrough infection, the immunogenicity of the SARS-CoV-2 variants, and the duration between the virus infection [51].

We observed a negative correlation between the antibody and T cell responses over time in the double dose vaccinated uninfected group, while the vaccinated post-COVID-19 group and homologous booster group demonstrated strong, positive correlation. In contrast, the heterologous booster group demonstrated weak correlation, highlighting the differential effects between different vaccination regimes. Nevertheless, our data support findings that T- and memory B-cell responses were maintained up to six months after infection, thus giving a stronger correlated response upon stimulation by the vaccine [26]. Although both antibody and T cell responses decline after initial peaks following vaccination [52,53], the presence of both immune components provides protection against infection [54] and from death in severe disease [55], which was also demonstrated in a macaque model [56]. As neutralizing antibodies tend to wane quickly [23,24], testing for T cell responses could provide more concise information on vaccine-generated immunity especially in long-term protection against SARS-CoV-2 and emerging variants, as opposed to determining immune protection based on antibody response alone [57]. However, our study demonstrated that neutralizing antibodies cannot be used as a predictor of T-cell immunity, and vice versa, as the correlation seems to vary according to how the immunity was achieved.

There are a few limitations in this study. Although it would have been interesting to also characterize immune responses in a cohort of people primed with BNT followed by ChAd booster, or those vaccinated with the BNT162b2 bivalent vaccine, such individuals were not available to us. The participants in this study, especially the booster cohort, were mostly healthy and relatively young individuals and therefore cannot represent elderly people or particular patient groups. Third, the QuantiFERON SARS-CoV-2 assay has a restricted time frame [58] and may not be able to detect T cell responses after a period of time (above 6 months) due to the reduction of effector T cell responses that could still be detectable by research assays [59,60]. Fourth, the small sample size, coupled with the short-term longitudinal study of six months, can be expanded to study the long-term durability of immune responses to SARS-CoV-2. 

## 5. Conclusions

In summary, we showed that both neutralizing antibody and T cell responses against SARS-CoV-2 could be detected for up to 3 months following the second and booster dose vaccine. The homologous booster regime confers a greater humoral response against SARS-CoV-2 WT, but similar cross-neutralizing antibodies against Omicron and T cell responses compared to the heterologous booster regime. The antibody and T cell responses are moderately correlated up to 6 months for the homologous booster and weakly correlated up to 3 months for the heterologous booster. While breakthrough infections in the homologous booster group significantly increased the level of neutralizing antibodies, T cell responses remained low. These data may impact government public health policies regarding the administration of mix-and-match vaccines, where both vaccination regimes can be employed should there be shortages of certain vaccines. 

## Figures and Tables

**Figure 1 viruses-15-00844-f001:**
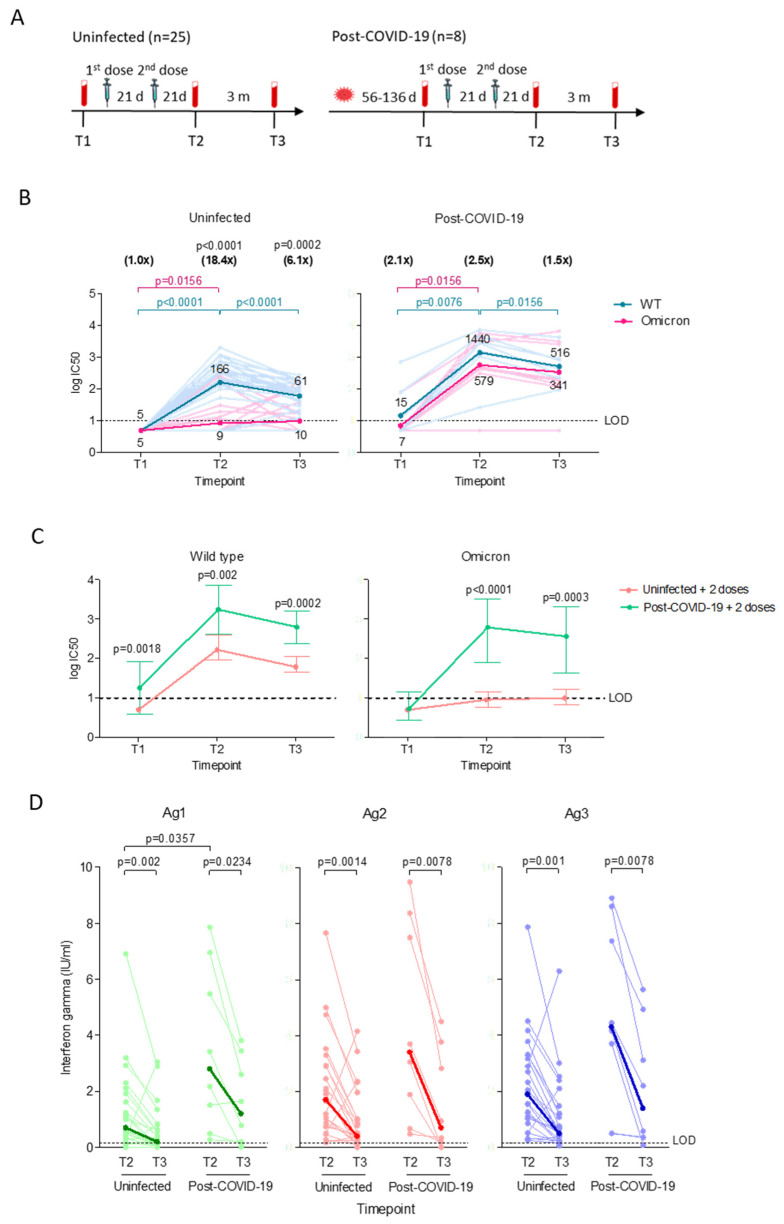
Neutralizing antibody and T cell immune responses after two doses of vaccines in uninfected (*n* = 25) and post-COVID-19 individuals (*n* = 8). (**A**) Longitudinal study design. Plasma samples were collected before the first vaccine dose (T1), and 21 days (T2) and 3 months (T3) after the second dose. (**B**) Neutralizing antibody titers (log IC_50_) against pseudoviruses bearing wild type (WT) or Omicron spike proteins. Significant differences within groups were assessed with the two-tailed Wilcoxon matched-pair signed rank test. The horizontal lines represent neutralizing antibody titers of individual samples against WT (light blue) and Omicron (light pink). The dark blue (WT) and dark pink (Omicron) lines represent the geometric means of log transformed values. P values in blue and pink represent significant differences in neutralizing antibody titers against WT and Omicron, respectively. The numbers in parentheses represent the GMT fold changes in neutralization against Omicron compared with WT and significant *p* values are stated above the fold change numbers. (**C**) Comparison of the GMT ± 95% confidence intervals against WT or Omicron. Significant differences between groups were assessed using the Mann–Whitney U-test. Significant p values for each comparison are shown. All experiments were performed in duplicate. (**D**) T cell immune responses against QuantiFeron SARS-CoV-2 Ag1, Ag2 and Ag3. Interferon-ɣ levels were normalized against the nil tube value (background). The Wilcoxon matched-pairs signed rank test was used to determine the significant differences within groups. Comparison between the uninfected and post-COVID-19 group at the same time-points were assessed using the Mann–Whitney U-test. Responses are shown as dot plots with connecting horizontal lines to represent interferon-ɣ changes in individual samples. The bold lines represent the medians of interferon-ɣ responses.

**Figure 2 viruses-15-00844-f002:**
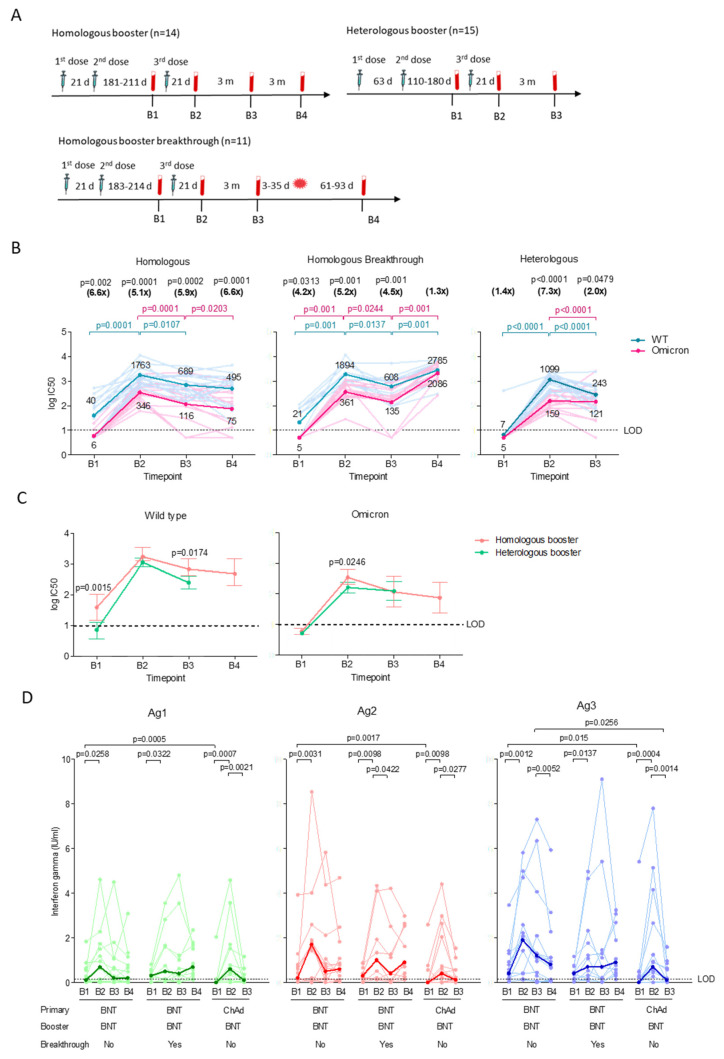
Neutralizing antibody and T cell responses of individuals who received a homologous (BNT-primed + BNT) (*n* = 14) booster and those who subsequently experienced breakthrough infection (*n* = 11), and heterologous (ChAd-primed + BNT) (*n* = 15) booster. (**A**) Longitudinal study design. Blood samples were collected before booster (B1), and 21 days (B2), 3 months (B3), and 6 months (B4) after the booster. For homologous booster breakthrough group, B4 sample was collected at 61–93 days after breakthrough infection. (**B**) Neutralizing antibody titers (log IC_50_) against pseudoviruses bearing wild type (WT) or Omicron spike proteins. Significant differences within groups were assessed with the two-tailed Wilcoxon matched-pair sum ranked test. The horizontal lines represent neutralizing antibody titers of individual samples collected at time-points against WT (light blue) and Omicron (light pink). The dark blue (WT) and dark pink (Omicron) lines represent the geometric means of log transformed values. *P* values in blue and pink represent significant differences in neutralizing antibody titers against WT and Omicron, respectively. The numbers in parentheses represent the GMT fold changes in neutralization against Omicron compared with WT and significant *p* values are stated above the fold change numbers. (**C**) Comparison of the GMT ± 95% confidence intervals against WT or Omicron spike. Significant differences between groups were assessed using the Mann–Whitney U-test. Significant *p* values for each comparison are shown. All experiments were performed in duplicate. (**D**) T cell immune responses against QuantiFeron SARS-CoV-2 Ag1, Ag2 and Ag3. Interferon-ɣ levels measured were normalized against the nil tube value (background). The Wilcoxon matched-pairs signed rank test was used to determine the significant differences within groups. Comparison between homologous booster, homologous booster breakthrough and heterologous booster groups at the same time-points were assessed using the Mann–Whitney U-test. Responses are shown as dot plots with connecting horizontal lines to represent interferon-ɣ changes in individual samples. The bold lines represent the medians of the interferon-ɣ responses.

## Data Availability

All data is contained within the article and Appendix A.

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
