# Peer review of "Humoral and T Cell Immune Responses against SARS-CoV-2 after Primary and Homologous or Heterologous Booster Vaccinations and Breakthrough Infection: A Longitudinal Cohort Study in Malaysia"

_viruses, 2023, doi:10.3390/v15040844_

Round 1

Reviewer 1 Report

    • Major comments: 

    In this study, Jolene Yin Ling Fu et al. did a longitudinal cohort study of 73 individuals in Malaysia to examine the humoral and T cell immune responses against SARS-CoV-2 after primary and homologous or heterologous booster vaccinations and breakthrough infection. They found that:

    1) vaccinated post-COVID-19 individuals showed higher neutralizing antibodies with more extended durability against SARS-CoV-2 wild type (WT) and Omicron spike but demonstrated similar declining T cell responses compared to uninfected vaccinated.

    2) two doses of BNT162b2 induced higher neutralizing antibodies against WT and T cell responses than ChAdOx1-S for six months.

    3) the BNT162b2 booster confers greater humoral response against WT, but similar cross-neutralizing antibody against Omicron and T cell responses in the homologous booster group compared to the heterologous booster group.

    4) Breakthrough infection in the homologous booster group (n=11) significantly increased the neutralizing antibody, but T cell responses remained low.

    The study is relevant to the field and well-organized.

    • General concept comments

    Here are some considerations/suggestions for the study:

    1.      The data of the titers of the serum collected anti-S and anti-N should be provided as supplementary data.

    2.      The authors should provide the rationale for why the study only relies on IFN-γ release over other cytokines to evaluate the T cell responses?

    3.      What are the actual protein sequences of Ag1 and Ag2, Ag3 also have the epitopes of the whole genome? Or the whole proteome? What SARS-CoV-2 strain do they belong to?

    4.      The T1 data of Ag1, Ag2, and Ag3 should be included in Figure 1D.

    • Specific comments:

    1)      Line 145, 1.5 × 105 relative luminescence units (RLU) of pseudovirus? How did you use RLU to quantify pseudovirus?

    2)      Line 208, GMT 344 here? It should be 341, according to Figure 1B.

    3)      Lines 218-219, it should be explained why the post-COVID-19 group is not showing significantly higher CD4+ and CD8+ responses against Ag2 and Ag3 compared to the uninfected group.

    4)      Lines 353-354, it should refer to Figure 3B here.

    5)      Figure 3C was not well illustrated.

    6)      Line 358, here P value was not consistent with Figure 3D.

    7)      Line 359, here P value was not consistent with Figure 3D.

    8)      Lines 427-430, well, it might also be because the prime and booster vaccines were based on wildtype SARS-CoV-2 strain rather than designed for Omicron strains, resulting in the low efficacy against Omicron pseudovirus in the study.

    9)      Lines 438-442, BNT vaccine encodes for the RBD of the spike S1 subunit? The authors should double-check the BNT vaccine information, and the explanations in Lines 440-442 could be wrong.

    10)  Lines 494-503, the updated bivalent vaccines, should be discussed here.

    11)  Line 508, but similar cross-neutralizing antibody responses?

Reviewer 2 Report

Your methods are clearly presented and your analysis is appropriate and clear.

However, your key problem is your very small sample. Of 100 young healthy individuals who were your eligibles you only studied 73 and divided them into five groups and computed differences: some groups were as small as 14 and one was 8. 

You have summarised your limitations very neatly:

"There are a few limitations in this study. Although it would have been interesting to 494 also characterize immune responses in a cohort of people primed with BNT, followed by 495 ChAd booster, such individuals were not available to us. The participants in this study, 496 especially the booster cohort, were mostly healthy and relatively young individuals and 497 therefore cannot represent elderly people or particular patient groups. Third, the Quanti- 498 FERON SARS-CoV-2 assay has a restricted time frame [58] and may not be able to detect 499 T cell responses after a period of time (above 6 months) due to the reduction of effector T 500 cell responses that could still be detectable by research assays [59-60]. Fourth, the small 501 sample size, coupled with the short-term longitudinal study of six months, can be ex- 502 panded to study the long-term durability of immune responses to SARS-CoV-2."

Therefore, I recommend that think this should be markedly shortened and resubmitted as a research note. 

Reviewer 3 Report

Overall, the study provides valuable insights into the humoral and T cell responses to SARS-CoV-2 vaccination in the Malaysian population and adds to the growing body of research on COVID-19 vaccination.

The article investigates the humoral and T cell responses against SARS-CoV-2 after primary and homologous or heterologous booster vaccinations and breakthrough infections in Malaysia. The study found that vaccinated post-COVID-19 individuals showed higher neutralizing antibody levels with longer durability against SARS-CoV-2 wild type (WT) and Omicron spike, but demonstrated similar declining T cell responses compared to uninfected vaccinated individuals. Two doses of BNT162b2 induced higher neutralizing antibody levels against WT and T cell responses than ChAdOx1-S for six months. The BNT162b2 booster conferred greater humoral response against WT, but similar cross-neutralizing antibody levels against Omicron and T cell responses in the homologous booster group compared to the heterologous booster group.

I have some minor comments:

The sample size of the study is relatively small. The study described in the Materials and Methods section follows a longitudinal design in which 100 individuals were recruited and divided into 5 groups: (1) 25 previously uninfected individuals receiving two primary doses of the BNT vaccine; (2) 8 post-COVID-19 individuals receiving two primary doses of vaccine; (3) 14 BNT-primed uninfected individuals receiving BNT booster (homologous booster group); (4) 15 ChAd-primed uninfected individuals receiving BNT booster (heterologous booster group); and (5) 11 individuals who experienced COVID-19 breakthrough infections after receiving three doses of BNT vaccines (homologous booster breakthrough group). The study design is appropriate for investigating the efficacy of different COVID-19 vaccination strategies. However, the number of participants in each group is relatively small, which may affect the statistical power of the analysis. Additionally, the study did not perform a power analysis on the number of individuals, which may lead to underpowered results.

The duration of follow-up is six months, which may not be enough to determine the long-term impact of booster vaccinations on humoral and T cell responses.

The methods used for plasma isolation and antibody detection are standard and well-established, and the results obtained from these assays can provide valuable information on the participants' immune response to COVID-19 vaccination.  The Interferon-γ release assay (IGRA) is a well-established method to measure T cell responses to antigens, including those induced by vaccination. In this study, SARS-CoV-2-specific T cell responses were evaluated using the QuantiFERON SARS-CoV-2 Starter and Extended Packs, which contain specific CD4+ and CD8+ T cell epitopes from the spike protein of the virus.

Overall, while the study design and methods used are generally appropriate, the small sample size and lack of power analysis may limit the generalizability of the findings.

Overall, the use of IGRA to evaluate T cell responses to SARS-CoV-2 vaccination is a valid and reliable method. The QuantiFERON SARS-CoV-2 Starter and Extended Packs provide a well-characterized set of T cell epitopes, allowing for consistent and accurate measurement of T cell responses to the virus. The statistical analysis used is also appropriate and comprehensive, covering a range of tests to determine differences between groups and within groups, as well as correlations between variables.

However, it is important to note that T cell responses are only one aspect of the immune response to vaccination, and it should be combined with other measures, such as antibody responses and clinical outcomes, to fully evaluate the effectiveness of the vaccine. Additionally, the study does not address the potential impact of variations in individual immune responses on the accuracy of the assay.

5. Conclusions

In summary, we showed that both neutralizing antibody and T cell responses against 505 SARS-CoV-2 could be detected for up to 3 months following the second and booster dose 506 vaccine. The homologous booster regime confers greater humoral response against SARS-507 CoV-2 WT, but similar cross-neutralizing antibody against Omicron and T cell responses 508 compared to the heterologous booster regime. The antibody and T cell responses are mod-509 erately correlated up to 6 months for homologous booster and weakly correlated up to 3 510 months for heterologous booster. While breakthrough infections in the homologous 511 booster group significantly increased the level of neutralizing antibody, T cell responses 512 remained low. These data may impact government public health policy regarding the ad-513 ministration of mix-and-match vaccines, where both vaccination regimes can be employed 514 should there be shortages of certain vaccines. Although the quantitative immune protec-515 tion level against SARS-CoV-2 infection remains unknown, and despite concerns regard-516 ing the loss of immunity against emerging SARS-CoV-2 variants [61-62], our study demon-517 strated that the booster dose remains relevant as immune memory responses persists and 518 are likely to protect against severe COVID-19 long term [63].

Comment about conclusion:

Overall, the study provides valuable insights into the duration and efficacy of both neutralizing antibody and T cell responses following vaccination with different booster regimes, which could inform public health policy decisions regarding the administration of vaccines during the COVID-19 pandemic. However, it should be noted that the study has some limitations, such as the relatively small sample size and short follow-up period, which may impact the generalizability of the findings. Further research with larger sample sizes and longer follow-up periods is needed to confirm and expand on these findings.

Pls note that references /citations are uncommonly used in Conclusion section.

Round 2

Reviewer 1 Report

I believe the manuscript has been improved, and the authors have addressed most of my concerns.

Author Response

Thank you for the comment.

Reviewer 2 Report

As before I was very concerned that this is a tiny study with groups as small as 8 and 11 persons and that reports could easily misrepresent the significance of the results. I advised reducing to a research note to capture the methods.

This has not been done.

Author Response

We understand that small sample size is a limitation in our study. We agreed to submit our study as a Brief Note as suggested by the reviewer.

In accordance with Brief Note format, we have revised the following:

  1. Changed Table 1 to Table S1 (supplementary).
  2. Combined Figure 2 (homologous and heterologous booster) and Figure 3 (homologous  booster breakthrough). Please refer to latest Figure 2.
  3. Shorten Result section.
  4. Removed Figure S3 and the initial Table S3.

We hope these changes are acceptable to the reviewer. Thank you.